

# Structure and functions of Yellow-breasted Boubou (*Laniarius atroflavus*) solos and duets

Amie Wheeldon[*], Paweł Szymański, Michał Budka and Tomasz S. Osiejuk[*]

Department of Behavioural Ecology, Institute of Environmental Biology, Faculty of Biology, Adam Mickiewicz University, Poznań, Poland
[*] These authors contributed equally to this work.

## ABSTRACT

**Background**. Birds have extremely well-developed acoustic communication and have become popular in bioacoustics research. The majority of studies on bird song have been conducted in the temperate zones where usually males of birds sing to attract females and defend territories. In over 360 bird species mostly inhabiting the tropics both males and females sing together in duets. Avian duets are usually formed when a male and female coordinate their songs. We focused on a species with relatively weakly coordinated duets, with male solo as the prevailing vocalisation type.

**Methods**. Instead of analysing a set of recordings spread over a long time, we analysed whole day microphone-array recordings of the Yellow-breasted Boubou (*Laniarius atroflavus*), a species endemic to West African montane rainforests. We described the structure of the solo and duet vocalisations and temporal characteristics of daily activity based on 5,934 vocal bouts of 18 focal pairs and their neighbours.

**Results**. Birds had small, sex specific repertoires. All males shared three types of loud whistles functioning as song type repertoires in both solos and duets. Females vocalised with five types of harsh, atonal notes with a more variable and usually lower amplitude. Three of them were produced both as solos and in duets, while two seem to function as alarm and excitement calls given almost exclusively as a solo. Solos were the most common vocalisation mode (75.4%), with males being more vocally active than females. Duets accounted for 24.6% of all vocalisations and in most cases were initiated by males (81%). The majority of duets were simple (85.1%) consisting of a single male and female song type, but altogether 38 unique duet combinations were described. Males usually initiated singing at dawn and for this used one particular song type more often than expected by chance. Male solo and duet activities peaked around dawn, while female solos were produced evenly throughout the day.

**Discussion**. Yellow-breasted Boubou is a duetting species in which males are much more vocal than females and duetting is not a dominating type of vocal activity. Duet structure, context and timing of daily production support the joint resource defence hypothesis and mate guarding/prevention hypotheses, however maintaining pair contact also seems to be important. This study provides for the first time the basic quantitative data describing calls, solos and duet songs in the Yellow-breasted Boubou.

Corresponding author
Tomasz S. Osiejuk,
osiejuk@amu.edu.pl,
t.s.osiejuk@life.pl

## INTRODUCTION

Birds have an extremely well-developed form of acoustic communication and so have become one of the most popular models in bioacoustics studies (*Catchpole & Slater, 2008*). The overwhelming majority of studies on bird song have been conducted in the temperate zone, a region in which only a small proportion of bird species breed (*Riebel et al., 2019*). Such research has been neglected in the tropics which is a region where extremely high bird biodiversity is observed (*Greenwood, 2001*). This geographical bias contributed to the formulation of an incorrect definition of bird song—an elaborate vocalisation produced by males during the breeding season to attract mates and defend territories (*Catchpole & Slater, 2008*). However, recently *Odom, Omland & Price (2015)* showed that female song performance is much more common than once thought, especially in the tropics, and was likely ancestral in oscine species (*Odom et al., 2014*). Although many songbirds are now classified as having female song, not many are truly monomorphic in singing (*Garamszegi et al., 2005*) with both male and female songs varying in acoustic rate, structure and complexity (*Price, 2015*). What is more, in many songbirds males and females sing together. Duetting can be described as vocalisations initiated by an individual that has a consistent time lag between the vocalisation of another individual, with this pattern being reproduced in the same way over time (*Langmore, 2002*). In summary, duetting is essentially a collective behaviour consisting of initiation by an individual and response of vocalisations by a different individual (*Logue & Krupp, 2016*). Duetting has been described for over 360 species across 50 different families of passerines and non-passerines (*Hall, 2009*). Despite the growing interest in female song and duets, and the vast reviews on functions (*Hall, 2009*), there can be further investigation for specific species, thereby adding to the general repertoire of knowledge being created (*Dahlin & Benedict, 2014*).

Research on duetting in birds has led to the formulation of many hypotheses trying to explain their functions. The historical development of these hypotheses was shown in a very detailed reviews by *Hall (2004)* and *Hall (2009)*. It was described that birds can gain different types of information through information conveyed to different receivers (partner, rivals or even predators) and this has evolved under cooperative or conflicting situations between partners (*Hall, 2004*). As a result, these hypotheses have diversified the potential for a general explanation of duetting evolution and functionality. Moreover, many different functions are not mutually exclusive and there may be a diverse use of duets in different species (*Hall, 2004*). For example, the sex recognition function (*Hooker & Hooker, 1969*) may in fact act as a pre-requisite for maintaining contact, synchronisation of reproduction, territory defence and so on. We do not want to repeat and discuss all the possibilities, but rather indicate the most promising explanations which seem to be important for the study species. Maintenance of contact between paired individuals using duets was found in habitats with dense vegetation and was one of the earliest functions proposed (e.g., *Thorpe & North, 1966*; *Lamprecht et al., 1985*; *Logue, 2005*). Duetting for maintenance of contact should occur all year round (*Odom et al., 2017*), whereas, if duets are used for reproductive synchrony, as suggested by *Dilger (1953)*, there should be a peak in activity around the nest building phase (e.g., *Topp & Mennill, 2008*). Mate

guarding—behaviour which clearly involves a conflict situation—can also utilise duetting behaviour. If an individual responds to its partner in a duet it advertises its mated status. Individuals can also answer a mate to guard paternity and deter rival males, as seen in male Rufous-and-white Wrens (*Thryophilus rufalbus*; *Kahn, Moser-Purdy & Mennill, 2018*). Another explanation related to a conflicting situation is signal jamming avoidance, which was experimentally shown to determine structure of duets in one of the pair-living antbird species (*Hypocnemis peruviana*; *Seddon & Tobias, 2009*). The other group of proposed duet explanations belongs to the joint resource defence hypothesis, which assumes that mated birds defend some resources, like a territory, together against outsiders (*Seibt & Wickler, 1977*). As with the aforementioned functions, one can expect a variety of male and female signalling strategies related to their locations, fighting abilities or mated status. For example, male and female birds may respond stronger to the same-sex intruder (*Logue, 2005*) or with equivalent intensity to both sexes as demonstrated in Barred Antshrikes (*Thamnophilus doliatus*) (*Koloff & Mennill, 2013*). Several hypotheses are more or less directly related to the fact that duetting behaviour is more often found in tropical birds than temperate species. Therefore, explanations for duetting are largely based on the differences between temperate and tropical birds' biology (*Hall, 2009*). Surprisingly, a recent broad-scale phylogenetic comparison revealed duetting evolved in association with the lack of migration, but not with sexual monomorphism or breeding in the tropics (*Logue & Hall, 2014*). Thus, despite the increasing number of studies on duetting birds, there is still a great need for basic duetting data for unexplored species.

One of the relatively well studied families of birds, with regards to duetting behaviour, are the bush-shrikes (Malaconotidae). These exclusively African birds are usually resident and are highly territorial, with a monogamous breeding system (*Harris & Franklin, 2010*; *Fry & Bonan, 2020*), thus evolved under ecological conditions promoting the evolution of coordinated defence of resources (*Logue & Hall, 2014*). Among bush-shrikes, the most abundant is the genus *Laniarius* with 22 species. The majority of *Laniarius* species are monomorphic in colour, with the exceptions differing slightly in colour with paler females, (*Harris & Franklin, 2010*) and utilise a skulking behaviour using their loud calls as the main sort of communication (*Sonnenschein & Reyer, 1984*). The Tropical Boubou (*Laniarius aethiopicus*), Crimson-breasted Gonolek (*Laniarius atroccineus*) and Slate-coloured Boubou (*Laniarius funebris*) have all been described as using duets for territorial defence and mutual mate guarding (*Grafe & Bitz, 2004*; *Van den Heuvel, Cherry & Klump, 2014*; *Sonnenschein & Reyer, 1984*). Although bush-shrikes have a relatively small acoustic repertoire they can alter the various parameters of their songs, such as the repetition of a note or the pitch, in order to produce more complicated or more simple duets (*Harris & Franklin, 2010*). *Grafe, Bitz & Wink (2004)* explain that the Tropical Boubou may have a more precise way of communication due to the large number of duet types in its repertoire, compared to other boubou species.

The formulation of hypotheses that allow for unambiguous testing of duet functions requires prior knowledge of natural song variation. Hence, the use of vocalisations obtained through natural, unprovoked settings provides a baseline for the standard behaviours and can be used as a guide for further experiments (*Mennill & Vehrencamp, 2008*). Therefore,

in the first step of our wider study we describe (1) the various vocalisation types produced by male and female Yellow-breasted Boubous (*Laniarius atroflavus*) and (2) we analyse how particular types of vocalizations are used as a solo or as part of a duet. Then, (3) we try to indicate the potential functions of the different vocalisation types based on natural vocalisation patterns. This is for vocalisations produced during within- and between-pair interactions. We used recordings from microphone arrays which allow for the analysis of whole day interactions between neighbouring pairs in the peak of the breeding season. The present study will increase the knowledge of duetting behaviours in a relatively well studied group of species. The addition of data on a new species adapted to living in a montane rainy forest (*Fry, 2020a*), should help to better understand factors affecting evolution of duetting in *Laniarius* species.

## MATERIALS & METHODS

The Yellow-breasted Boubou is a sexually monochromatic (likely human perception only, see *Osinubi et al., 2018*) and socially monogamous bush-shrike, endemic to the montane forests of south-eastern Nigeria and western Cameroon (*Stuart, 1986*; *Borrow & Demey, 2001*; *Fry, 2020a*). Pairs inhabit dense undergrowth at the edge of clearings, secondary scrubs, small forest remnants and bamboo highlands above 1500 m above sea level (*Riegert, Přibylová & Sedláček, 2004*) in which they hold year-round territories. On Mount Cameroon Yellow-breasted Boubous can also be found at lower elevations, from 700 m above sea level (*Fry, 2020a*). While there has been a description of Yellow-breasted Boubou vocalisations (*Riegert, Přibylová & Sedláček, 2004*; *Fry, 2020a*), little is known about their function or the context in which they are produced. It merely informs us about how they sound rather than what the vocalisations mean or in what context they are produced. Both males and females produce solos as well as initiate duet bouts. They are vocally active throughout the year, but with a clear peak that starts at the beginning of the dry season (late November - January; *Olszowiak, 2018*; P Szymański et al., 2020, unpublished data).

### Study area and population

Our study area was located in the Bamenda Highlands, near to Big Babanki village in the Northwest Region of Cameroon (6°5′–6°8′N and 10° 17′–10°20′E). The study area was covered by approximately 12 km$^2$ of montane habitat (from 1,900 to 2,400 m above sea level). The Bamenda Highland region is one of the most important hotspots of bird diversity and endemism in Africa but, due to intensive logging in recent decades, the formerly continuous forests have been reduced to isolated patches (*Orme et al., 2005*; *Reif et al., 2006*). During the study, the habitats within the study area were a mosaic of montane forest patches, shrubby corridors and grasslands, with vegetable plantations below 1,800 m above sea level. The study species was common in this area and was found inside larger forest patches, as well as in smaller remnants along streams. In these areas its population was continuous, and the Yellow-breasted Boubou vocalizations were one of the most common signals heard (*Reif et al., 2006*). Detailed characteristics of the habitats in the study area is presented in *Budka et al. (2020)*.

The study was conducted at the beginning of the dry season (November-December), a time when most bird species in this region start to breed (*Serle, 1981*; *Tye, 1992*; *Sedláček et al., 2007*). During this period, we observed boubous building nests, laying and incubating eggs, and adult birds with young. Our own observations suggest that the breeding period may start at the beginning of November but that it can be elongated, as brood losses are quite common and so pairs may attempt to breed multiple times.

## Microphone array recordings

In 2014 (from 12 November to 5 December) we recorded birds with eight automatic recorders (Song Meter SM3 connected with dedicated GPS receivers; Wildlife Acoustics) organised into a microphone array. Recorders were put on trees in such a way that their microphones had active ranges covering the territories of up to three focal pairs whilst also recording their adjacent neighbours. From our own recordings that cover a 24 h period, we know that this species only incidentally produce vocalisations at night (M Budka et al., 2010–2011, unpublished reconnaissance material; 16 points recorded continuously 48 hrs with SM2 Wildlife Acoustics song meters in 2010). Therefore, all recorders were synchronised ($\pm$ 1 ms accuracy) by the GPS in such a way that they started recording at 05:00 (sunrises were between 06:06 and 06:14) and stopped recording at 19:00 (sunsets were between 17:58 and 18:01). This recording regime allowed us to obtain the entire vocal activity of the Yellow-breasted Boubou pairs. SM3 units recorded single channel soundscape with 48 kHz frequency sampling and 16 bits quality. Altogether we collected array recordings for 18 focal pairs, covering eight separate areas, producing a whole day activity recording using an eight-channel microphone array setup (see Fig. S1). In each of the eight areas we recorded 1-3 focal pairs bordered with 1 or 2 recognised neighbours. We used an 8-channel microphone array to simultaneously record 3 pairs in 3 sessions, 2 pairs in 4 sessions and 1 focal pair in 1 session. These numbers reflected natural locations and sizes of particular territories and made it possible to place the microphones in a specific way so we could assign a particular channel(s) to a particular pair, based on the highest amplitude. If focal pairs produced vocalisations, they were always recorded on three or more channels within the microphone array.

## Definitions used for describing vocalisations and sound analysis

Bird vocalisations are traditionally divided into songs and calls, and songs are usually louder and longer than calls and are involved in mate attraction and territory ownership (*Catchpole & Slater, 2008*). However, Yellow-breasted Boubous produce a variety of vocal signals which are short, relatively simple and are not intuitively easy to classify into one of these two separate categories. Based on scarce literature data, our own preliminary observations and recordings, we tried to use song and call terms, together with the naming of vocalisations based on their structure in an onomatopoeic way (referring when possible to *Fry, 2020a*). In further terminology, describing vocalisations and duets in particular, we try to apply suggestions presented by *Hall (2009)* and *Logue & Krupp (2016)* (see Fig. 1):

  –call – short and simple vocalisation, usually used in specific contexts such as alarm, begging; etc.;

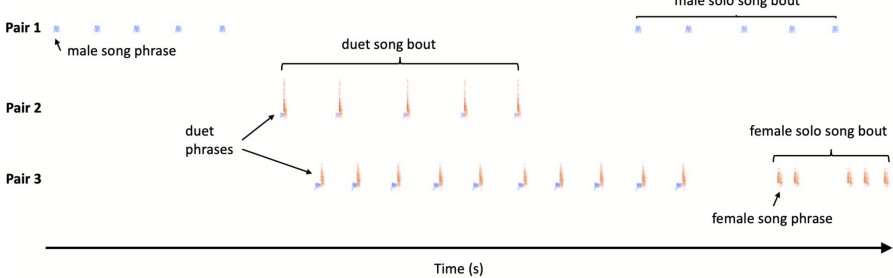

**Figure 1** **Sketch illustration of multichannel recording output with different pairs of the Yellow-breasted Boubous producing alternating, overlapping and type matching song bouts.** For simplicity only three channels presented.

–song – vocalisations used for advertising mate or territory ownership;

–phrase – unit within a song, which may be an element (uninterrupted trace on sonogram) or set of elements occurring together;

–call bout and song bout – continuous call or song phrase output, where calls or phrases are separated by a silent interval (gap) lasting substantially longer than intervals between calls or phrases within the bout;

–call type and phrase type – version of call or song phrase, which could be defined on the basis of a specific (repeatable among individuals) structure;

–duet – coordinated singing by male and female so that their phrases alternate or overlap; in the study species duets usually consist of two or more phrases and form a duet bout, the equivalent of a 'duet train' like male–female-male–female etc. (*Brown & Lemon, 1979*);

–duet type –particular combination of the phrase types used by duetting birds;

–solo –song bout consisting of a single or a series of phrases produced in a sequence by one individual and separated from its other vocalisations by a substantially longer time than intervals within the bout; for the study species the same phrase types were used for solos and duets and so our definition of solo is equivalent to that proposed by *Logue & Krupp (2016)* which is the initiation of a duet which remained without answer.

Each call and song bout can be characterised by its: duration (s), number of units (calls or phrases) produced by a male, female or both sexes and rate (units / min). For duets one may also calculate sex bias –defined here as a ratio of female to male phrases in a single duet bout. Sex bias reflects the contribution of a particular sex to a duet train (*Logue & Krupp, 2016*).

Recognition of individuals. Assignment of vocalisations recorded of particular pairs was a multi-step process. First, one person (AW) assigned each vocalisation bout to a particular song or call type category and to a particular pair (or non-focal neighbour) based on the highest amplitude on a particular channel (see Fig. S1). A simple map showing locations of each recorder (and respective channel on multi-channel file) in relation to a territories position was used as an aid. For the majority of cases there was no problem as birds called from known positions within their territories and usually for a short time of a few seconds or for a few minutes (depending on the time during the day), with neighbours responding

from their own positions. Birds from outside the recorded area appeared on a single (edge) channel and were easy to recognize as non-focal birds, due to the low amplitude presented on the array channels (see Fig. S1). In addition, a second person (TSO) was checking all identified bouts and in case of any doubts was checking, in detail, characteristics of a particular bout. Doubts usually appeared because of the quality of songs, e.g., when target sound overlapped with signals of other species. With male song phrases it was easy, despite a fully shared repertoire, to assign particular individuals as each male song from a particular category has its individual specificity reflected by small but consistent differences in frequency and duration. This time-frequency characteristic of male calls was already used in a methodological study on measuring individual identity in general (*Linhart et al., 2019*). We compared the shape of phrases with the Peak Frequency Contour measurement of Raven Pro with measurements visible on screen and listening to the signal at a slow speed. In case of doubts because of quality we also used measurements of frequency and time in order to compare phrases directly with earlier recordings of recognised males (Fig. 2). Please notice, that for each session we only need to discriminate between a maximum of three focal males and 1-2 additional neighbours (assigned only to the category non-focal pair). To our knowledge it is not possible to discriminate between females based on their simple time-frequency characteristics of songs (personal observations). Therefore, for female solos we assigned bouts solely based on the location in which they were produced. In such cases the preceding and following bouts of neighbours or their own partner, make such assignments certain. Hence, the main potential error in our dataset may be a result of singing by focal females from outside of their own territories and assignment of such bouts to other pairs or non-focal birds. However, based on our observations of colour-ringed birds we assume that such cases, if any, were extremely rare (personal observations).

Sound analyses were done in Raven Pro v. 1.5 (Cornell Lab of Ornithology, Ithaca, NY; http://www.birds.cornell.edu/raven). All eight channels of the microphone array recordings were visually inspected (with auditory examination if necessary) and all call and song bouts were selected within the channel with the highest signal amplitude which came from the recorder placed in the song activity centre of a particular territory. Additional annotation columns were added to each recording in a standard way and, as a consequence, each selection containing a bout included the following information: time of the start and end (actual and in relation to sunrise and sunset time), category of bout (call, solo or duet), sex of initiator, type and number of units produced by each sex, pair identity (based on location and individual call characteristics) and additional notes.

At this stage all calling and singing bouts were selected from recordings and the following parameters of Raven Pro were used: Window type: Hann, 1,024 samples; 3 dB Filter Bandwidth: 67.4 Hz; Time grid: overlap 50% giving Hop Size: 512 samples; Frequency Grid: DFT Size: 1,024 samples giving 46.9 Hz $\times$ 10.7 ms resolution of measurements.

## Statistical analysis

To quantitatively characterise the production of male and female solos, as well as duets, basic descriptive statistics were used. We focused on the frequency of different vocalisation bouts produced by focal pairs, and quantified them by the number of phrases, duration and
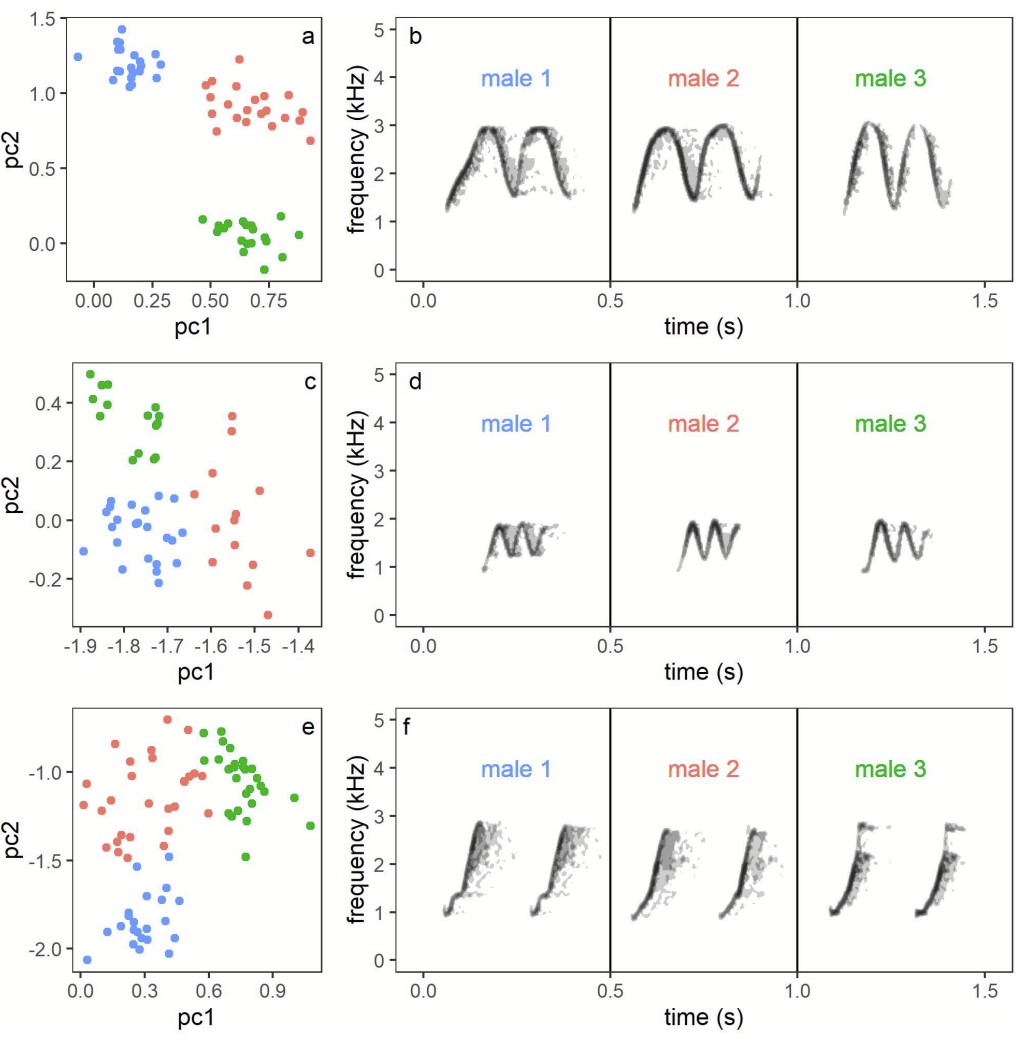

**Figure 2 Illustration of individual variation male song phrases (B, D, F), which enables individual recognition (A, C, E).** (B, D, F) present three examples of each song type derived from the repertoire of three males: (B) High whee-oo, (D) Low whee-oo, and (F) Hwee-hwee. What is visible at first glance, despite sharing the same general pattern within a song phrase type, each individual has some specificity in the course of song whistles. These differences are easy to detect when Peak Frequency Contour measurement of Raven Pro is chosen with option Enable Measurement Plots on. Moreover, these small differences are individually invariable as illustrated in (A, C, E). Scatterplots illustrated with colour separation are of male song phrases for several song phrases measured for three males. Variables pc1 and pc2 are the two first principal components derived from original time-frequency variables of song phrase variation measured in Raven Pro (Principal Component Analysis with Varimax rotation and Kaiser normalisation). The pc1 explained 48.9% of original song variation and had the stronger loadings on several frequency measures (like low and high frequency, delta frequency, Q1 and Q3 frequency and peak frequency); the pc2 explained 37.9% of original variation and had the heaviest loading on time related measures (e.g., delta time, IQR Duration).

rate. In order to characterise the general daily pattern of vocalising we counted the number of different bout classes (e.g., call bouts, solos, duets etc.) produced by each pair during every hour of activity and with reference to the time of sunrise. In addition to descriptive

statistics, we used generalized mixed models (GLMM) with a log-link function and Poisson error distribution, or identity link function and Gaussian error distribution, that included pair identity as a random factor, with time of day (hour in relation to sunrise), type of bout (solo, duet), sex (male, female) and duet initiator (male or female) as explanatory effects. All statistical analyses were performed using the program STATA/MP 16.x (StataCorp, College Station, Texas, USA). Mean $\bar{x} \pm$ SE values and 95% CI are reported.

### Ethical approval

This study was exclusively observational, and due to national law for this type of study formal consent is not required (The Act on Experiments on Animals (Disposition no. 289 from 2005). However, it was part of a wider project which as a whole was approved by the Local Ethical Committee for Scientific Experiments on Animals permit no. 16/2015, and Polish Laboratory Animal Science Association 1952/2015 certificate to TSO.

## RESULTS

### Sound material analysed

In total, eight whole day recording sessions were analysed with 1-3 focal pairs recorded simultaneously ($N = 18$ pairs). This produced nearly 900 hrs of single channel recordings in which we found 5,934 call and song bouts which contained a total of 88,442 calls and song phrases. Among those bouts, 4,753 (80%) were assigned to the 18 focal pairs, while 1,181 (20%) were considered as being produced by neighbours from adjacent territories outside the microphone array based on their appearance on particular channels of the recording.

### Types of call and song bouts produced

As many as 75.4% of all bouts recorded ($N = 4,472$) were produced by a male (63.2%) or by a female (36.8%). However, among female solos 991 bouts (16.7% of all bouts) were call bouts or non-song vocalisations (more details below); duets accounted for 24.6% ($N = 1,462$) of all bouts. We found that the phrases of males and females produced in solos and duets were easy to categorise to a limited number of classes based on audio detection and visual inspection of spectrograms.

### Male solos

Males produced three whistle phrase types named High whee-oo, Low whee-oo and Hwee-hwee (Fig. 3). We found very consistent and statistically significant differences in proportions of these three phrase types used by all males as solos (GLMM, $\beta \pm$ SE = $-0.21 \pm 0.014$; $z = -14.75$, $p < 0.001$). High whee-oo were produced the most often $57 \pm 2.3\%$ (95%CI [52.2–61.4]%), then Low whee-oo $28 \pm 1.8\%$ (95%CI [24.3–31.5]%), and Hwee-hwee $15 \pm 1.2\%$ (95%CI [12.8–17.8]%).

### Female solos

Female vocalisations had a completely different acoustic structure, being atonal, harsh notes of differing durations. Most of them were classified as Keck (59.1%), Chock-series (32.5%), Chock (3.8%) and Kee-roo (3.5%) with very few examples of Rasp (1.1%) (Fig. 4).

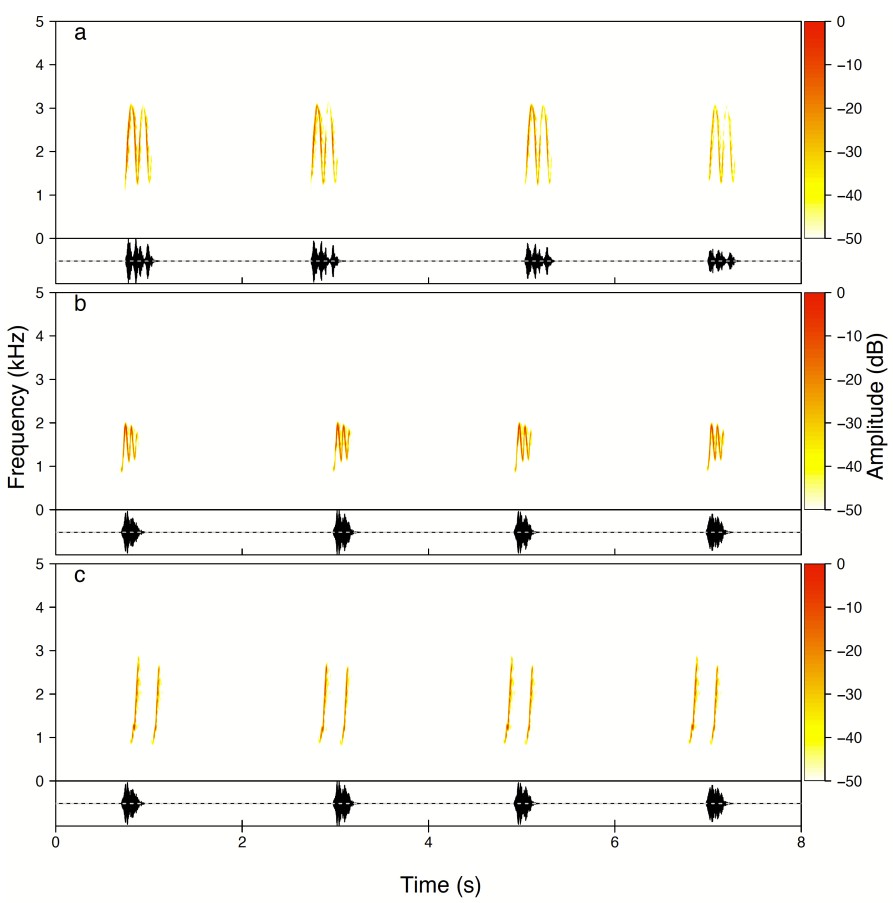

**Figure 3** **Examples of all three types of the Yellow-breasted Boubou male songs produced as solo.** (A) High whee-oo, (B) Low whee-oo, and (C) Hwee-hwee phrase types. Singing rate is typical for the species. The relevant sounds are included as Supplemental Information.

Kecks were rattle-like calls exclusively produced in a high rate series consisting of up to hundreds of single and very short notes. Visual observations clearly suggest that Kecks were produced in an alarm context, e.g., close to the nest. Chock-series were always produced as a series of 2–14 calls with a high rate, almost without gaps between phrases (0.8–0.15 s) and with up to 14 phrases in a row. Chocks had a similar but distinguishably different structure to chock-series, and were produced as a single, double or triple-unit as one phrase after another but without consistent spacing in time, apparently different to the characteristics for the Chock-series. Rasps were very rarely produced (recorded only 33 times) and to our knowledge they are given in the context of high excitation (personal observations). Rasps were also relatively quieter in comparison to the other vocalisations, and because of that might, on occasion, have not been recorded. Therefore, we did not include them in most of the analyses. Based on both array recordings and observations of vocalising birds we cannot state that Kecks and Rasp calls are also produced by males.

Unlike males, the proportions of phrase types (Chock-series, Chocks and Kee-roos) used for solo singing were very variable (GLMM, $\beta \pm SE = -0.29 \pm 0.054$; $z = -5.44$, $p < 0.001$)

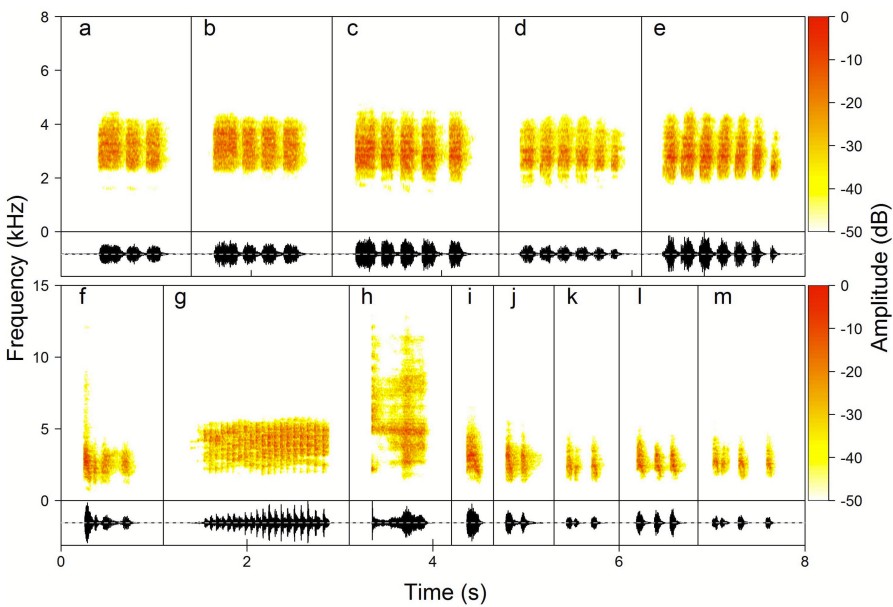

**Figure 4 Examples of the Yellow-breasted Boubou female song phrase and call type repertoire.** (A–E) Chock-series phrases always produced in series, here from 3 to 7, and used by females to initiate duets; (F) Kee-roo song phrase, (G) Keck—alarm call, (H) Rasp—excitation call, (I–M) Chock song phrases used in response to male and to lead duet. The relevant sounds are included as Supplemental Information.

among females from the 18 focal pairs. Chock-series were the most commonly observed call type among female solos: $71 \pm 8.7\%$ (95% CI [53.8–88.0]%), then Kecks $16.8 \pm 8.2\%$ (95% CI [0.3–33.4]%), Kee-roos $12.2 \pm 4.3\%$ (95% CI [3.1–20.7]%) and finally Chocks $5.6 \pm 2.7\%$ (95% CI [0.01–10.98]%). If we consider female notes classified as functional calls, Kecks were commonly used by all females $98.1 \pm 0.01\%$ (95% CI [96.6–99.7]%) while Rasps were found $1.9 \pm 0.7\%$ (95% CI [0.33–3.39]%) incidentally.

## Duets

Yellow-breasted Boubous used the same phrase types for duetting as were used for solo vocalisations. Among 1,462 analysed duetting bouts, 85.1% were simple duets consisting of a single type of both male and female phrases. In 81% of cases duets were initiated by males and in 19% by females, however, even if a female initiated a duet, she usually reverted to following the male components of a duet. Even in duets where female phrases prevail over male phrases, female phrases were organised in time in relation to male elements which were always produced with a very constant rate (Fig. 5). The most typical male initiated duets used High whee-oo phrases (52%) then Low whee-oo (38%) and finally the Hwee-hwee phrase type (10%). Female initiated duets most often used the Chock-series (42%), Kee-roo (35%) and Chock (20%) phrase types (Table S1).

When we focused on duets produced by the 18 focal pairs, we found that only one duet type was found in the repertoire of all pairs. It was initiated by females using the Chock-series, then males produced the High whee-oo and females overlapped these phrases with Kee-roo. Another few duet types which were common and found in the

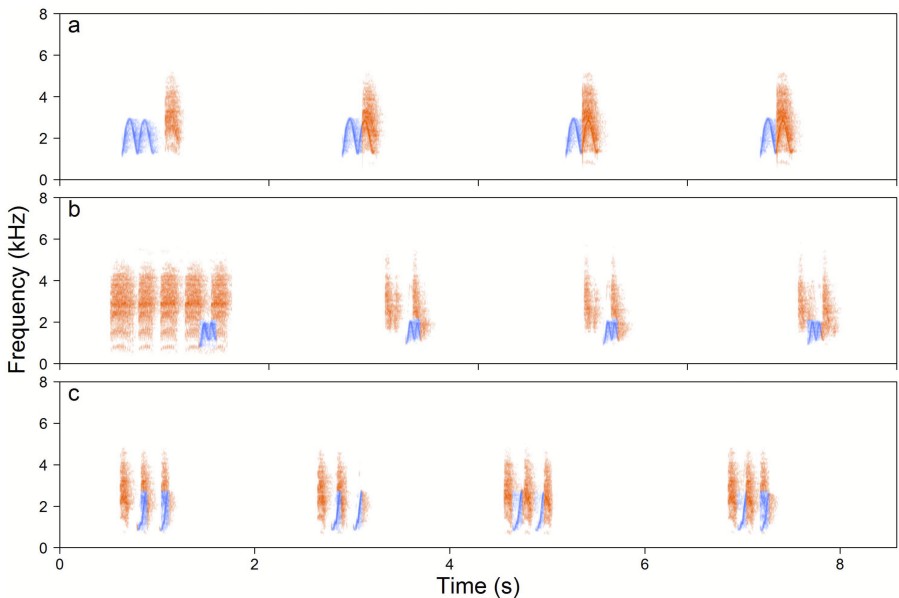

**Figure 5** **Examples of the Yellow-breasted Boubou duets.** (A) male-initiated and male-led duet—male High whee-oo and female Chock, (B) female-led duet—female Chocks and male Low whee-oo, and (C) female-initiated and female-led duet—triple or double or female Chocks and male Hwee-hwee phrase types. Duetting rate is typical for the species. The relevant sounds are included as Supplemental Information.

repertoires of the majority of pairs were also simple in structure and consisted of a single male and female phrase produced with a time overlap. We found 16 duet types produced only once by a single pair and their uniqueness was that in a single duetting bout male and/or female switched between different phrase types. More details are in Table S1.

## Duet initiation and answering analysis

If we assume that every spontaneous song phrase produced by a male or female has been answered by its mate, we may consider that our results reflect individual decisions (*Logue & Krupp, 2016*). In total, the study species tend to sing more in solos than duets. The three male phrase types remained unanswered by the female in 60.8–79.4% of cases (Table S1). The very common female phrase Chock-series remained unanswered by a mate in 81.9% of cases (Table S1). The female phrase types Kee-roo and Chock remained unanswered in 37.0% and 53.8% of cases, respectively. A completely different pattern was found for Keck calls as they were almost never answered (99.8%) by males. The Rasp calls were also rarely answered by males (24.0%), but they were also very rarely recorded. We speculate that both Kecks and Rasps are not produced by females to form duets, but just when females are alarming (Kecks) or are highly excited (Rasps), males may also produce song phrases, but not in a coordinated way with the female (Table S1).

## Temporal characteristics of male and female solos

Male solo bouts produced phrases with surprisingly similar average rates (Table 1) which did not differ significantly between phrase types (GLMM, $\beta \pm SE = 0.02 \pm 0.795$, $z = 0.03$,

**Table 1 Temporal characteristics of song phrases and call bouts of Yellow-breasted boubou produced solo by males and females.**

| Vocalization | No of units in a bout | | Bout duration (s) | | Rate (units/min) | |
|---|---|---|---|---|---|---|
| | $\bar{x} \pm$ SE | 95% CI | $\bar{x} \pm$ SE | 95% CI | $\bar{x} \pm$ SE | 95% CI |
| Male solos | | | | | | |
| High hwee-oo | 8.0 ± 0.20 | 7.66–8.45 | 26.0 ± 0.73 | 24.60–27.49 | 35.1 ± 0.69 | 33.72–36.43 |
| Low hwee-oo | 10.0 ± 0.34 | 9.31–10.64 | 31.5 ± 1.17 | 29.20 ± 33.81 | 32.5 ± 1.01 | 30.49–34.44 |
| Hwee-hwee | 11.1 ± 0.68 | 9.81–12.47 | 31.4 ± 2.04 | 27.39–35.41 | 36.9 ± 1.27 | 34.38–39.38 |
| Female solos | | | | | | |
| Chock | 6.8 ± 0.59 | 5.63–8.01 | 43.6 ± 3.19 | 37.25–50.02 | 94.9 ± 43.07 | 8.80–181.07 |
| Chock-series | 5.0 ± 0.07 | 4.84–5.10 | 1.9 ± 0.01 | 1.25–1.31 | 172.4 ± 18.12 | 136.82–208.05 |
| Kee-roo | 5.7 ± 0.91 | 3.93–7.57 | 8.54 ± 2.11 | 4.33–12.76 | 114.9 ± 10.31 | 94.23–135.57 |
| Keck[a] | 28 ± 01.43 | 25.22–30.86 | 8.72 ± 0.42 | 7.89–9.55 | 232.1 ± 4.28 | 223.71–240.52 |
| Rasp[a] | 4.4 ± 0.73 | 2.84–5.89 | 18.79 ± 6.22 | 5.71–31.86 | 64.8 ± 10.28 | 43.25–86.44 |

Notes.
[a]Keck and Rasp vocalizations were recognized as functional calls (alarm and excitement). See text for details.

$p = 0.976$). On the other hand, the differences in the number of phrases within a bout (GLMM, $\beta \pm$ SE $= 0.07 \pm 0.012$, $z = 5.92$, $p < 0.001$) and as a consequence the bout duration (GLMM, $\beta \pm$ SE $= 0.05 \pm 0.015$, $z = 3.62$, $p < 0.001$) were significantly different between bouts produced with different phrase types (with the following pattern High whee-oo >Low whee-oo >Hwee-hwee). Thus, males produced solos with a very regular and fixed rate, but obviously changed bout duration by producing more or fewer phrases in a series. We did not record male solo bouts with more than a single phrase type.

A different situation was found for females (Table 1). As was mentioned already, three types of vocalisations (Chock-series, Chocks and Kee-roos) were used by females as songs, while the remaining two were used as calls (Kecks and Rasp). In the majority of cases the Chock-series remained unanswered by males and they were never repeated one after another. Chocks and Kee-roos produced as a solo had similar temporal organisation, typically with 4-8 notes in a bout (Table 1) and they were used both to initiate duets and as a response to males during duets. Female solo song bouts of different types (Chock-series, Chocks and Kee-roos) significantly differed in number of phrases (GLMM, $\beta \pm$ SE $= 1.04 \pm 0.204$, $z = 5.07$, $p < 0.001$), duration (GLMM, $\beta \pm$ SE $= 2.81 \pm 0.192$, $z = 14.63$, $p < 0.001$) and call rate (GLMM, $\beta \pm$ SE $= -76.21 \pm 3.901$, $z = -19.53$, $p < 0.001$). Keck calls were clearly different from other vocalisations, as they were produced with extremely high rates and sometimes in a very long series (Table 1). Rasps were recorded rarely, hence it is hard to temporally characterise them in more detail. However, recorded examples indicate a sudden and irregular appearance (Table 1).

## Temporal characteristics of duets

On average, birds used 22.3 ± 0.63 (95%CI [21.1–23.5]) phrases in a duet, and the average duet duration was 30.3 ± 0.89 s (95%CI [28.5–32.0]). The rate of duet phrases doubled those of solos, with an average of 68.0 ± 1.27 phrases per minute (95%CI [65.5–70.5]). Duets initiated by males were longer (on average 32.0 vs 24.7 s; GLMM, $\beta \pm$ SE $= 4.81 \pm 2.042$, $z = 2.35$, $p = 0.019$), but contained fewer phrases (21.4 vs 25.1;

GLMM, $\beta \pm SE = -3.57 \pm 1.721$, $z = -2.07$, $p = 0.038$) and had a lower rate (62.5 vs 94.6 phrases/min; GLMM, $\beta \pm SE = -29.22 \pm 3.229$, $z = -9.05$, $p < 0.001$) than female initiated duets.

We found significant differences in the number of the male and the female phrases in duets initiated by male and female (GLMM, $\beta \pm SE = -1.75 \pm 0.114$, $z = -15.41$, $p < 0.001$). If duets were initiated by males, the number of male and female phrases within a duet was almost equal (Sex bias $= 0.98 \pm 0.014$, 95%CI [0.95–1.01]). However, if females were initiating duets, they produced significantly more phrases than males (Sex bias $= 2.67 \pm 0.206$, 95%CI [2.27–3.08]). Characteristically, males responded to females initiating duets with any type of their song phrase repertoire (Table S1). If a female initiated the duet with a Chock-series she always switched later in a bout to Chock or Kee-roo phrases (e.g., Fig. 4B, Table S1). Consequently, Chock-series were never used within a duet and never repeated one after another.

Diurnal pattern of calling activity during breeding season. We found that Yellow-breasted Boubous started to vocalise on average $16 \pm 6.1$ mins before sunrise (95%CI [30.0–3.3] mins before sunrise; extremes from 61.1 mins before to 23.7 mins after sunrise) and that singing activity was the highest during the first two hours after sunrise (Figs. 6–7). Interestingly, birds were vocally active during the whole day, even between 11:00 and 15:00 when the temperature was usually quite high (24.8–31.0 °C) in comparison to dawn (14.5–16.4 °C; P Szymański et al., 2020, unpublished data). Characteristically, the number of bouts per hour in which females were involved were small (Figs. 6–7), and we found no significant trends for number of female solos produced during the daytime (GLMM, $\beta \pm SE = 0.07 \pm 0.047$, $z = 1.54$, $p = 0.124$) and duet bouts initiated by females (GLMM, $\beta \pm SE = -0.02 \pm 0.019$, $z = -0.85$, $p = 0.393$). Thus, the main part of the overall variability of the singing activity during the day resulted from the activity of male solos and duets initiated by males (Figs. 6–7). The number of male song bouts significantly decreased during the day time (GLMM, $\beta \pm SE = -0.32 \pm 0.107$, $z = -2.95$, $p = 0.003$), although male initiated duets did not differ significantly throughout the day (GLMM, $\beta \pm SE = -0.10 \pm 0.057$, $z = -1.78$, $p = 0.075$).

We analysed who, and with what call type, first started vocalising in the morning. When we analysed 18 focal pairs, 78% of cases started with a male solo bout (and 9 of these 14 cases were males calling with the Hwee-hwee phrase type). Duets were observed as the first call bout in two pairs (11%; Kee-roo —Low whee-oo and High whee-oo —Kee-roo) as were female solos (two cases, 11% of Kecks). A long series of Kecks given by females were observed (personal observations) as an apparent response to a threat (human or squirrels close to nest) and so these two early cases of Kecks given by females might be interpreted as an unspontaneous dawn chorus but are more likely used as a response to a predator.

## DISCUSSION

Here we provide the first paper to thoroughly explore the form and potential functions of the vocalisations, both in solo and duet form, of the Yellow-breasted Boubou. Through the use of a microphone array setup we have been able to analyse natural singing and
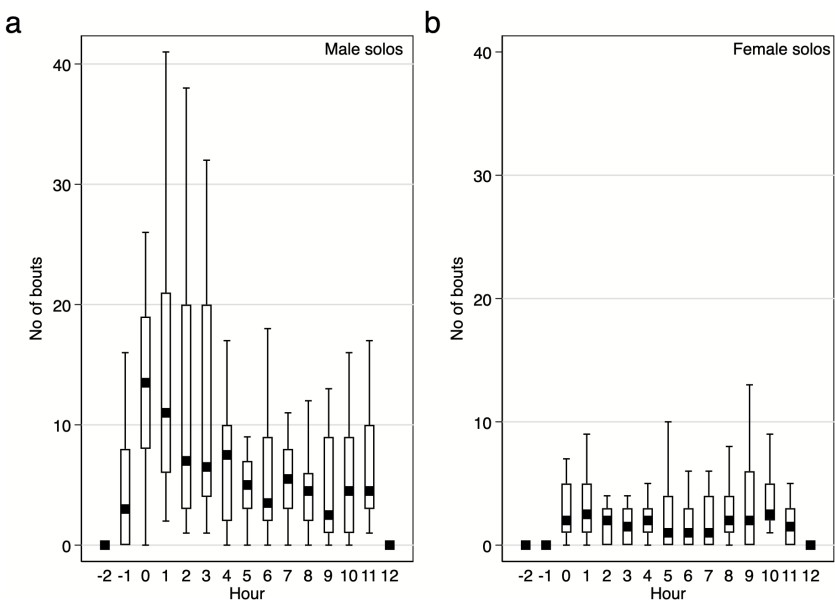

**Figure 6  Number of solo song bouts of males and females per hour / per pair of the studied Yellow-breasted Boubou.** Boxes indicate median, 25th–75th percentile and lower-upper adjacent values. Only data for 18 focal pairs were included.

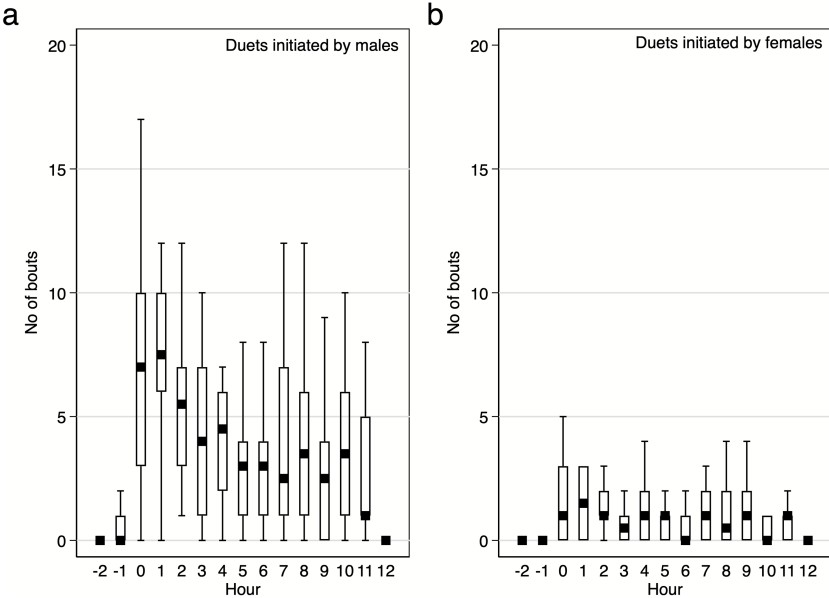

**Figure 7  Number of male and female initiated duet bouts per hour / per pair of the studied Yellow-breasted Boubou.** Boxes indicate median, 25th–75th percentile and lower-upper adjacent values. Only data for 18 focal pairs were included.

calling behaviours of this species which provides basic information about the study species' vocalisations and can later be used to better interpret experimentally induced behaviours.

## Repertoire of male and female vocalizations

We found that the Yellow-breasted Boubou has a small and sex specific repertoire of vocalisations that are usually used in both a solo and duet context. The males perform three distinct, tonal song phrases whilst the females vocalise with five atonal, harsh notes (Figs. 2–3). All three male song types were produced with a very repeatable pattern characterised by a fixed repetition rate both between phrase types and between males. Moreover, series of tonal songs produced by males were similar in solos and duets.

For females we recorded five types of sexually specific, atonal vocalisations, but only three of them seem to be functional song units (Chock-series, Chocks and Kee-roos). They were produced as solos or in duets, and when they were performed together with male vocalisations they were coordinated with the male output (Fig. 5). The common Keck call was given by females almost exclusively as a solo. Based on our visual observations this is an alarm call produced in the context of the potential presence of a predator, e.g., squirrel close to the nest or a human. The Rasp calls were recorded extremely rarely and if they appeared alongside a male call (14 bouts only) they were not synchronized precisely in time. Again, visual observations of such displays were found in most of the cases during playback experiments and suggest that Rasp is a high excitation call (A Wheldon et al., 2020, unpublished data testing response of focal pairs to different types of song). For example, they were recorded during a failed experiment where two adjacent pairs approached the speakers and met and chased each other aggressively (personal observation). To summarise, within the studied population males and females used sexually dimorphic vocalisations, both in the context of solo and duet singing.

## Duets

The rules of duet organisation for the Yellow-breasted Boubou seem to be simple: (1) both sexes may start a duet but males do so much more frequently than females; (2) males always produce their phrases with a very regular time pattern while females add one or more phrases (Chock or Kee-roo) per single male call; (3) males and females initiated duets with any kind of their sex specific song phrases, but female Chock-series were never produced inside duets; (4) the majority of duets consist of one male and one female phrase type only; (5) females produce more phrases per male phrase if they started the duetting bout. So, the duets of the study species are sex specific, and male and female components are easy to identify even from a longer distance. As summarised in *Hall*'s (*2004*) review, loud, locatable and sex-specific duet elements support the hypothesis for maintaining contact. Indeed, in the Yellow-breasted Boubou the environment is visually occluded and the duet could be initiated in order to locate a partner. On the other hand, they also produce duets when sitting right next to each other and so the maintaining contact hypothesis is not the only function (*Hall, 2004*). The aforementioned duet properties are also described as being linked to guarding/preventing partner usurpation as well as for joint resource defence (*Hall, 2004*) and it seems Yellow-breasted Boubous also duet to convey information about their

mated status and defence ability. Further research on this topic demands more detailed information about the duet characteristics in relation to duet context and the status of each bird.

When assessing duet function, it is important to look at the different sexes. Yellow-breasted Boubous have a sex specific repertoire used for both solo and duet bouts. Within the *Laniarius* genus, the situation of sex specific call types is complicated. For example, in some species like the Gabela Bush-shrike (*Laniarius amboimensis*) and Red-naped Bush-shrike (*Laniarius ruficeps*), males and females produce structurally similar phrases (*Fry, 2020b*; *Fry, 2020c*). However, the Tropical Boubou has strictly sex specific phrases when performing duets (*Grafe & Bitz, 2004*), with males using tonal whistles and females producing both (sex specific) tonal whistles and atonal notes. Another interesting bush-shrike is the Southern Boubou (*Laniarius ferrugineus*) in which males and females exchange phrase types when producing duets (*Wickler & Seibt, 1982*). With the absence of plumage or size dimorphism in certain duetting species, the ability to produce sex specific song types is one way that duet members can establish mate guarding or paternity guarding behaviours through sex recognition (*Hall, 2004*). Both the Tropical Boubou and Crimson-breasted Shrike (*Laniarius atrococcineus*) are Malaconotidae species that utilise sex specific songs for mate guarding behaviours (*Grafe & Bitz, 2004*; *Van den Heuvel, Cherry & Klump, 2014*), and so it is likely that the role of sex specific songs in the Yellow-breasted Boubou is a function of mate guarding behaviour in this monomorphic species. If we compare all *Laniarius* species for which we have any data on vocal behaviour (*Winkler, Billerman & Lovette, 2020*), it seems that in the majority of cases males tend to produce whistle like phrases while females use (at least more often) atonal harsh notes. Such differences may suggest some functions which remain to be studied in detail. Tonal whistles are more efficiently propagated through dense forest habitat (*Boncoraglio & Saino, 2007*) which, together with a higher amplitude, suggests that male phrases (A Wheeldon et al., 2020, unpublished data) are aimed at receivers at a further distance than the phrases produced by females. Although males share all phrase types they are clearly individually distinct (*Linhart et al., 2019*). At the moment we do not know if this is also the case for females due to the complexity of the atonal harsh notes and limited amount of isolated female recordings in the field. However, experiments suggest that females can discriminate easily between their own mate and stranger males based on songs while there is no evidence that it works the other way around (P Szymański et al., 2020, unpublished data). Such observations support the idea that differentiated structures of male and female song only reflect their functional distinctiveness.

## Diurnal pattern of vocal activity

Knowledge of the temporal pattern of singing is another aspect of duetting behaviour necessary to understand its function. In this study we collected material representing a pairs' activity for an entire daily activity period during the peak of the breeding season. The analysed material was collected during eight different days between 12 Nov and 5 Dec, and for 18 pairs, hence it is rather unlikely that it is biased because of, for example, unusual weather or random events (e.g., losing brood). In general, the Yellow-breasted Boubou has

a classic diurnal pattern of vocal activity, with a clear peak early in the morning and smaller peak in activity before dusk. Hence, this pattern was similar to that of other duetting species (e.g., White-eared Ground sparrow *Melozone leucotis*, *Sandoval, Mendez & Mennill, 2016*). Several more detailed observations may help in linking their solos and duets with particular functions. For example, Yellow-breasted Boubous do not exhibit any regular diel variation in any of the duet types used and peaks of diurnal activities were largely caused by male solos or any duets initiated by males. Similarly, the Tropical Boubou which produces up to 12 duet types did not exhibit any consistent variation of how these types are used during the day (*Grafe & Bitz, 2004*).

For the Yellow-breasted Boubou, differences in durations of male and female unanswered solos suggest that males are regularly producing long bouts of solos, which are often responded to by neighbouring males (or pairs), whilst females are just trying to evoke a male response and stop calling shortly after if there is no response. Thus, we observed some kind of dichotomy of vocal activity for males and females. The only female vocalisation type to show any diel variation was the Keck call which is produced more often at the end of the day and with a high calling rate. It seems that this call type is linked to an alarm context as it was often produced when human observers were close, and usually followed by the males' appearance (personal observations). *Langmore (1998)* explains that certain female call types may be used to coordinate the care of young, and so it may be that such vocalisations are produced by the females of the study species in order to synchronise certain behaviours with their mate. Therefore, we do not rule out that this alarm call can also be used to summon the mate.

Females of the study species vocalise less than males in both solo and their initiated duets. The amount in which females sing in the tropics varies across species. Chirruping Wedgebill (*Psophodes cristatus*) females vocalise at a lower rate than males (*Austin et al., 2019*), this reduced rate of vocalising could be because females may only increase the amount of singing if a mate dies and so they need to be able to hold a territory independently (*Langmore, 1998*). Conversely, in certain species the females have an increased singing activity compared to males. Slate-coloured Boubou (*Laniarius funebris*) females have a higher vocal activity due to aggressive encounters (*Wickler & Seibt, 1979*). In general, it seems that females singing more intensively than males are relatively rare. *Dutour & Ridley (2020)* indicated only six such species in literature, and some of them concern duetting birds, for example the Cocos Flycatcher (*Nesotriccus ridgwayi*) (*Kroodsma et al., 1987*) and New Zealand Bellbird (*Anthornis melanura*) (*Brunton & Li, 2006*; *Brunton et al., 2008*). A reason for the variability in male and female vocalisation rates may be due to the hormonal balance in a species, with higher testosterone levels equating to increased vocal activity (*Odom et al., 2014*). It appears the Yellow-breasted Boubou males are more vocally active than females as there is less need for aggressive solo displays by females and possibly a lack in intense female–female competition due to the monogamous life history strategy pursued. However, this interpretation must be treated with caution, as it is known that in some closely related species, despite social monogamy the proportion of extra pair offspring could be substantial (*Van den Heuvel, Cherry & Klump, 2014*).

The dawn chorus acts as a communication network, whether signals are directed at an individual or are eavesdropped by other individuals (*Burt & Vehrencamp, 2005*). In the Yellow-breasted Boubou, the first calls at dawn are typically produced by males as solo calls, followed by female solos and duets. Surprisingly, the least frequently produced male phrase type, the Hwee-hwee, was usually used as the first vocalisation type in the morning. In Banded Wrens (*Thryophilus pleurostictus*), vocalisations that are produced in the dawn chorus are usually longer and have a higher bandwidth than other songs in their repertoire (*Trillo & Vehrencamp, 2005*). However, the Hwee-hwee phrase is not used exclusively as an early morning song and is not so structurally different from other male whistles. Yellow-breasted Boubou pairs hold stable, year-round territories and so it seems that the morning peak in male vocal activity followed by the females joining mates in duets might have a double function. It could be interpreted as something like checking the attendance list, which could be important for both within-pair as well as between-neighbour communication. Similarly in White-eared Ground-sparrows, solos are produced as the first vocalisation type as a way of demonstrating pair bond maintenance (*Sandoval, Mendez & Mennill, 2016*) however, it may also be a way of eliciting extra-pair copulations. Black-capped Chickadee (*Poecile atricapillus*) females can compare the solo songs sung by males in the morning and use this to assess fitness (*Gammon, 2004*). Yellow-breasted Boubou pairs are described as utilising a monogamous breeding system (*Harris & Franklin, 2010*) and so it is likely that the first solos calls produced are a means of pair-bond maintenance or territorial defence, rather than to seek extra-pair paternity opportunities.

## CONCLUSIONS

Yellow-breasted Boubous represent a duetting species in which males are more vocally active than females and duetting is not a dominating type of vocal activity. Males and females have distinctive, small and sex specific repertoires used both in solos and duets. There is a dawn chorus effect shown with male solos that can be interpreted as a form of within and between pair communication. We found some interesting differences in vocalisation types used for both males and females, suggesting that some songs and calls may have specific functions. Our findings suggest that male solos and duets initiated by males are used for territorial defence. On the other hand, the female singing pattern with more effort being put into female-initiated duets suggests that their own calls are directed to own mates.

## ACKNOWLEDGEMENTS

We thank Ernest Vunan Amohlon for his help in organising field work in Cameroon and all Kedjom-Keku People Community for allowing to study birds on their land, and dr Moses Njoya from Bamenda University for help in organising local permits.

### Funding

This work was supported by the Polish National Science Centre under Grant UMO-2015/17/B/NZ8/02347 to Tomasz S. Osiejuk. The funders had no role in study design, data collection and analysis, decision to publish, or preparation of the manuscript.

### Grant Disclosures

The following grant information was disclosed by the authors:
The Polish National Science Centre: UMO-2015/17/B/NZ8/02347.

### Competing Interests

The authors declare there are no competing interests.

### Author Contributions

- Amie Wheeldon and Tomasz S. Osiejuk conceived and designed the experiments, performed the experiments, analyzed the data, prepared figures and/or tables, authored or reviewed drafts of the paper, and approved the final draft.
- Paweł Szymański conceived and designed the experiments, performed the experiments, prepared figures and/or tables, authored or reviewed drafts of the paper, and approved the final draft.
- Michał Budka conceived and designed the experiments, performed the experiments, authored or reviewed drafts of the paper, and approved the final draft.

### Data Availability

Raw data for all recorded and analyzed song bouts are available in the Supplemental Files.

### Supplemental Information

Supplemental information for this article can be found online at http://dx.doi.org/10.7717/peerj.10214#supplemental-information.

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
