# Peer review of "Structure and functions of Yellow-breasted Boubou (Laniarius atroflavus) solos and duets"

_PeerJ, doi:10.7717/peerj.10214_

## Round 0.1 · original submission · Minor Revisions

For the most part this is a nice descriptive study; I agree with reviewer 2 that this type of study is important and appreciate this one. My main concern is that reviewer 1 feels that that methods need to be a lot clearer. Please follow all suggestions to do so. I agree with the comments of reviewer 2 regarding the length of the discussion (too long) and that some of the comparisons with others species in that section seem tangential.

Please fix the following small errors.

line 19 - “A majority”
line 42 - “study provides”
line 43 - “solos”
line 44 - I’m not following what the phrase “what allows for planning” is doing in this sentence.
line 99 - change “the recent” to “a recent”
line 133 - cut “The difference being” and replace “this” in the next line with “which”
line 234 - change “Fristly” to “First”
line 569 - missing an “in” after “Similarly"

Reviewer 1 ·

Basic reporting

The paper is in general good writing; some sentences need more explanation (see comments). The paper is focused on a particular genus and the majority of the information is bias in that direction. I will recommend that authors use examples from other species in other tropical regions that study duets and diel patterns for example. The paper provides new important information for the group of investigators that study boubous, about duets as mentioned before, the discussion is biased to this group, so this makes this manuscript poorly attractive to other duet investigators.

Experimental design

My main concern is with the method of multichannel recording. Authors need to explain more how they recognize each pair and individual. For example, they can provide actual Raven screen captures, where they signaling how they separated continuous pairs unambiguously. Additionally, they did not explain well how they solve the ambiguous ID and how that data was included in the manuscript.

Validity of the findings

In general, the manuscript is well writing and results look interesting and well analyzed and presented.

Additional comments

Line 48: add a reference
Line 50: add a reference
Line62-63: this sentence is unclear, what are you trying to summarize
Line65-66: Although the information in this sentence is true, as is presented looks like that we do not know anything about duets. This appreciation is unfair, because a lot of work has been made to understand the functions, and we know very well some functions and codes. So, I recommend changing how this is presented to avoid intruding on a mistake in the duet knowledge.
Line 129-131: How using a microphone array you can describe male and female vocalizations and the context use?
Line 133-135: It is clear for you, but not for a reader. Please explain better this sentence.
Line 135-137: Why this information about the other boubou species is important?
Line 138-141: After read your introduction the information in this sentence is unclear. You need to explain more your idea that a small distribution range will affect duet behavior. Additionally, without a phylogenetic tree or information is hard to understand if your selected species is the base of the group or is more recent in origin. I say this because it is important to understand the phylogenetic relationship to disentangle the habitat effect from species distribution.
Line 154: change sols for solos
Line 242-243: You need to explain more in detail how the information in this sentence. What are all recognitions? When occurred a doubt? What this means: was checking, in detail, characteristics of a particular bout?
Line 243-246: You need to provide more information about this method, with detail enough for a reader that never read the Linhart et al., 2019 paper, could understand how you recognize male vocalizations in your recordings
Line 248: What is a doubt?
Line 252-255: Unclear
Line 257-258: How common this occurs, please add a number of observations, and how many times you observed this occurring?
Line 434: what is the meaning of: Calls or songs?
Line 438: what is a stable repetition rate? Add more information about what is a repetition rate
Line 440-441: Songs are very diverse in bird species. So, if you defined song as: “vocalisations used for advertising mate or territory ownership”, why to say that “male phrase types are the equivalent of song types”. This makes that a reader thinks that your song definition did not work. Delete this sentence.
Line 449-452. You mention here visual observation but in methods is little or nothing about how much visual observation was conducted to associate vocalizations with behavior.
Line 456-564: These sentences are results
Line 490-492: This sentence is unnecessary for your discussion because did not provide support or discussion to your idea of the sex role in vocalizations.
Line 502-504: Add more information about why you suggest this.
Line 509-515: Very speculative paragraph without support information. Delete it.
Line 525-527: Something that may help you to understand the diel patterns is to refer to other tropical species studies where was analyzed the diel pattern for solo and duet songs.
Line 544: You need to provide examples of species where females vocalize more than males, it is important the contrast.
Line 551: Could be that the female vocalizes more in aggressive context because this reduces the probability that her male will be stolen for another female, as has been proposed in other species?
Line 556: They are truly monogamous or have extra-pair copulations? How EPC will affect the territorial behavior between females?

·

Basic reporting

Some minor copy editing is needed to correct typos and improve the written English, but overall the basic reporting is clear and sufficient.

Raw data are provided. It would also be useful if example sound files were included as supplementary material so that readers could listen to the different vocalizations.

Experimental design

This is a descriptive study, rather than an experiment. The methods and analyses are appropriate. Data such as these are critical to facilitate future comparative studies.

The only shortcoming is in Figure 2. Plots of PC analyses used to differentiate songs from different individuals are provided, but no details on these analyses are provided other than a very short descriptor on line 249.
If the PC analysis is described in another study I would remove the PC plots and just leave the spectrograms. Alternatively, the methods for the PC analysis could be added as supplementary information.

Validity of the findings

Meets journal standards.

Additional comments

Kroodsma and Byers (1991) wrote "to experiment first is human, to describe first, divine." Descriptive studies such as this one are the foundation on which experimental and hypothesis-testing work can be conducted. Thus I congratulate the authors on providing a thorough and detailed description of vocalizations for this duetting bird species.

The article is generally clear. I think the discussion could be shortened. There is quite a bit of comparison with other boubou species that can be removed to make this paper more focussed, and comparisons across species could be the topic of a comparative or review paper.

Minor comments:
line 121: it is not clear what "species-adequate" means
line 144: monochromatic to human visual systems, not necessarily to birds
l. 143: change sols to solo
l 351 change sinfle to single
l 364, the interpretation of these calls indicating alarm or excitement do not have any evidence provided. These interpretations should be removed, or clearly indicated as speculation.
l. 424 how were these apparent threats observed based on songmeter recordings?
l. 426 remove extraneous text

---

## Round 0.2 · accepted · Accept

Thank you very much for your thorough revisions, which in particular have made the methods easier to follow.

Reviewer 1 ·

Basic reporting

The authors addressed correctly all my coments

Experimental design

The methods improved with the changes conducted

Validity of the findings

The manuscript is still very focused in a genus of birds, which will reduce the manuscript impact